# Changes in Free-Living Glycemic Profiles after 12 Months of Lifestyle Intervention in Children with Overweight and with Obesity

**DOI:** 10.3390/nu12051228

**Published:** 2020-04-26

**Authors:** Kylie Karnebeek, Jesse M. Rijks, Elke Dorenbos, Willem-Jan M. Gerver, Jogchum Plat, Anita C. E. Vreugdenhil

**Affiliations:** 1Centre for Overweight Adolescent and Children’s Healthcare (COACH), Department of Paediatrics, Maastricht University Medical Centre, 6229 HX Maastricht, The Netherlands; kylie.karnebeek@mumc.nl (K.K.); jmrijks@gmail.com (J.M.R.); elke.dorenbos@mumc.nl (E.D.); 2School of Nutrition and Translational Research in Metabolism (NUTRIM), Maastricht University, 6229 ER Maastricht, The Netherlands; 3Department of Paediatrics, Maastricht University Medical Centre, 6229 HX Maastricht, The Netherlands; w.gerver@maastrichtuniversity.nl; 4Department of Nutrition and Movement Sciences, School of Nutrition and Translational Research in Metabolism (NUTRIM), Maastricht University, 6229 ER Maastricht, The Netherlands; j.plat@maastrichtuniversity.nl

**Keywords:** childhood obesity, glucose metabolism, lifestyle intervention, continuous glucose measurement

## Abstract

Previous studies demonstrated that hyperglycemic glucose concentrations are observed in children that are overweight or have obesity. The aim of this study was to evaluate the effect of a 12 month lifestyle intervention on free-living glycemic profiles in children that were overweight or had obesity, and the association of the alterations with changes in cardiovascular risk parameters. BMI z-score, free-living glycemic profiles, continuous overlapping net glycemic action (CONGA), and cardiovascular parameters were evaluated before and after a multidisciplinary lifestyle intervention, in 33 non-diabetic children that were overweight or had obesity. In children with a decrease in BMI z-score, the duration which glucose concentrations were above the high-normal threshold (6.7 mmol/L) and the glycemic variability decreased significantly. In these children, a decrease in median sensor glucose was associated with decreases in LDL-cholesterol, and systolic and diastolic blood pressure z-score. A decrease in BMI z-score was associated with a decrease in CONGA1, 2, and 4. In conclusion, the glycemic profiles in free-living conditions in children that were overweight improved in children with a decrease in BMI z-score after lifestyle intervention. In those children, changes in median sensor glucose concentrations were associated with changes in LDL-cholesterol and blood pressure z-scores. These results suggest that glucose homeostasis can improve after one year of lifestyle intervention and that these improvements are associated with improvements in cardiovascular health parameters.

## 1. Introduction

It is well acknowledged that children that are overweight or have (morbid) obesity are at risk for developing type 2 diabetes mellitus (T2DM) and cardiovascular diseases (CVD) [1,2]. There are strong suggestions that mild glycemic dysregulation, which precedes the actual onset of T2DM, contributes substantially to the development of endothelial dysfunction [3,4]. Several studies have shown aberrant cardiometabolic risk profiles, including dyslipidemia and hypertension, at a young age in children with overweight and with (morbid) obesity [5,6,7]. These early cardiometabolic aberrations in childhood were shown to be strong predictors of the development of atherosclerotic cardiovascular disease in adulthood [8].

In a recent study, we demonstrated that, besides the presence of an increased CVD risk, glucose homeostasis is already disturbed in these children [9]. Hypoglycemic and hyperglycemic excursions (respectively in 73% and 27% of the children) were frequently observed in children with overweight and with (morbid) obesity, using a continuous glucose monitor (CGM) sensor in free-living conditions [9]. In that study, 91% of the glucose measurements were within the normoglycemic range (3.9–7.8 mmol/L). Chan et al. also demonstrated hyperglycemic excursions in free-living conditions in adolescents with obesity and pre-diabetes [10]. Another previous study showed that hypoglycemia and hyperglycemia were less frequent in children with a normal weight, with hypoglycemic excursions in 24% of the children and 97% of the glucose measurements being within the normoglycemic range [9]. Altogether, these findings suggest that the vascular system of children with overweight and with (morbid) obesity is already exposed to glycemic dysregulation at an early age. This exposure is likely to be harmful, and indeed, the duration and magnitude of hyperglycemic glucose excursions were demonstrated to be associated with cardiovascular risk parameters such as triacylglycerol concentrations and waist circumference in children with overweight and with (morbid) obesity [9]. Moreover, in healthy adults and adults with T2DM, a high frequency and amplitude of glucose fluctuations during the day (high glycemic variability) initiated oxidative stress pathways and pro-inflammatory cytokine secretion, both having harmful effects on vascular function [11,12,13,14]. Moreover, besides affecting the peripheral vasculature, a high glycemic variability was shown to have a negative effect on brain development in children with type 1 diabetes [15].

In adults with T2DM, glucose disturbances were shown to be reversible, since lifestyle interventions improving dietary behavior or physical activity resulted in a significant improvement of glycemic variability in free-living conditions [16,17,18,19]. Current studies investigating continuous glucose monitoring in free-living conditions in children focused mainly on the use of these measurements in individuals with diabetes [20,21,22]. Studies investigating glycemic profiles in free-living conditions in non-diabetic children are scarce, limited to cross-sectional evaluations, and the effects of lifestyle improvement on glycemic profiles in free-living conditions are unknown [9,10]. Furthermore, whether the improvement of glucose homeostasis due to lifestyle changes translates to cardiovascular health benefits in children with overweight and with (morbid) obesity remains to be explored. Therefore, the aim of this study was to evaluate the effect of 12 months of lifestyle intervention on glycemic profiles in children with overweight and with (morbid) obesity in free-living conditions, and to evaluate the association of alterations in these glycemic profiles with changes in cardiovascular risk parameters.

## 2. Materials and Methods

### 2.1. Setting

This study was designed and conducted within the setting of the Centre for Overweight Adolescent and Children’s Healthcare (COACH) at the Maastricht University Medical Centre (MUMC+). Within COACH, the health status of children with overweight and with (morbid) obesity and their families was evaluated; they were monitored and received lifestyle coaching as described previously [5]. Briefly, participation in the COACH program started with a comprehensive assessment aimed at excluding underlying syndromic or endocrine conditions leading to being overweight, evaluating complications and risk factors, and obtaining insight into dietary and physical activity behavior and family functioning. The assessment included, amongst others, a CGM sensor measurement and an oral glucose tolerance test (OGTT). After the assessment, all children and their families were offered frequent, on-going, tailored, and individual guidance at the outpatient clinic, focusing on durable lifestyle changes. Focus points for the intervention were identified in the initial assessment. These focus points could mainly be divided into the following categories: nutrition (e.g., adequate fruit and vegetable intake), food habits (e.g., portion size), physical activity (e.g., limiting sedentary time and increasing exercise), sleep (e.g., sleep hygiene and duration), and psychological and social aspects (e.g., self-esteem and bullying). Although all of these focus points were evaluated in all children and their families, the emphasis could differ based on individual needs. Focus points were continuously reassessed throughout the intervention period and adapted if necessary. The goal of the intervention was not to quickly lose weight, but to make small, stepwise changes towards healthy lifestyle behavior and convert these into daily habits that could be maintained in the future.

Furthermore, participation in sports activities in groups and activities aimed at increasing nutritional knowledge were offered. A follow-up assessment including all of the examinations performed during the initial assessment was offered annually to all children.

### 2.2. Study Participants

All 43 children with complete CGM sensor data at baseline and who had additional CGM sensor measurements after 12 months of intervention were considered for inclusion in this study. Children with incomplete CGM sensor data after 12 months of intervention were excluded from this study (*n* = 10). Finally, 33 children were eligible for inclusion. Informed consent was obtained from all subjects and/or their parents. The study was conducted according to the guidelines administered by the Declaration of Helsinki and approved by the medical ethical committee of the MUMC+. It is registered at ClinicalTrial.gov as NCT02091544.

### 2.3. Participant Characteristics

Anthropometric measurements were acquired while children were barefoot and wearing only underwear. Body weight was determined using a digital scale (Seca), and body length was measured using a digital stadiometer (De Grood Metaaltechniek). Body mass index (BMI) was calculated and BMI z-scores were obtained using a growth analyzer (Growth Analyser VE), based upon reference charts of the Dutch nationwide growth study [23]. Based on the International Obesity Task Force criteria, age and sex-dependent cut-off values were used to classify children as being overweight (comparable to a BMI of 25 kg/m^2^ in adults), having obesity (comparable to a BMI of 30 kg/m^2^ in adults), or having morbid obesity (comparable to a BMI of 35 kg/m^2^ in adults) [24]. Waist circumference was measured with a non-elastic tape, at the end of a natural breath, at the midpoint between the top of the iliac crest and the lower margin of the last palpable rib. Waist circumference z-scores were calculated according to age references for Dutch children [25]. Ethnicity was defined based on the definition of the Dutch Central Agency for Statistics [26].

### 2.4. Glucose Metabolism

Fasting plasma glucose concentrations (spectrophotometry, Cobas 8000 modular analyzer, Roche), serum insulin concentrations (Luminescence immuno enzymatic assay, Immulite-1000, Siemens Healthcare Diagnostics), and HbA1c concentrations (high-performance liquid chromatography, HPLC Variant II, Bio-Rad Laboratories) were determined. After obtaining the fasting blood sample, an OGTT was performed. An amount of 1.75 grams of glucose per kilogram of bodyweight was dissolved into 200 mL water, with a maximum of 75 grams of glucose in total, and given orally. Plasma blood glucose concentrations were measured every thirty minutes over two hours. Impaired fasting glucose (IFG; fasting glucose 5.6–6.9 mmol/L), IGT (≥7.8 glucose <11.1 mmol/L after 2 hours), T2DM (fasting glucose ≥7.0 mmol/L or glucose ≥11.1 mmol/L after 2 hours), and elevated HbA1c concentrations (≥5.7%) were defined according to the American Diabetes Association (ADA) criteria [27]. In this study, insulin resistance was estimated using the HOMA-IR [28]. The following formula was applied: fasting plasma glucose (mmol/L) x fasting serum insulin (µU/L) / 22.5 [28]. A cut-off point of 2.5 was used, based on adult standards, to determine the presence of insulin resistance [28]. 

Glucose concentrations in free-living conditions were measured using a CGM sensor (MiniMed, Metronic), as described previously [9]. In short, sensor glucose concentrations were measured in the interstitial fluid for 48 hours in free-living conditions, and median, minimum, and maximum glucose concentrations were calculated. The durations that glucose concentrations were in the hypoglycemic ranges (level 1: 3.9–3.0 mmol/L; level 2: <3.0 mmol/L), the target range (3.9–10.0 mmol/L; secondary 3.9–7.8 mmol/L), and hyperglycemic ranges (level 1: >10.0 mmol/L; level 2: 13.9 mmol/L) were calculated according to the International Consensus on Use of Continuous Glucose Monitoring [29]. Additionally, since the children that were included in this study did not have diabetes and were therefore unlikely to have glucose concentrations in the aforementioned hyperglycemic level 1 and 2 ranges, but could be expected to have high normal or mildly elevated glucose concentrations, the durations of glucose concentrations being ≥6.7 mmol/L and ≥7.8 mmol/L [29,30] were also calculated. The sensor glucose AUC was calculated using the trapezoidal method. Overall glycemic variability during the whole 48-hour period was assessed using the coefficient of variation (CV), which was calculated as: (standard deviation/mean) × 100%. The sensor glucose measurements were also stratified for daytime and nighttime glucose measurements (day: 7 a.m.–10 p.m.; night: 10 p.m.–7 a.m.). Furthermore, the intra-day glycemic variability, which reflects acute glucose fluctuations throughout the day, was assessed with the continuous overlapping net glycemic action (CONGA). With this method, the difference between each glucose concentration and the glucose concentration at a time point certain hours previously is calculated [31]. The CONGA is the standard deviation of the differences. In this study, CONGA1, CONGA2, and CONGA4 were used based on 1-, 2-, and 4-hour time differences, respectively. 

### 2.5. Cardiovascular Risk Parameters

Fasting lipid and lipoprotein profiles, including serum total cholesterol (TC), serum LDL-cholesterol (LDL-C), serum HDL-cholesterol (HDL-C), and serum triglyceride concentrations, were measured (by spectrophotometry with the Cobas 8000 modular analyzer, Roche). Daytime blood pressure (BP) was measured during a period of 1.5 hours—approximately 20 times, with an interval of three minutes between each measurement—using the Mobil-O-Graph (I.E.M. GmbH). Based on these 20 measurements, the mean BP was calculated. The size of the cuff depended on the circumference of the upper arm. Systolic and diastolic BP *z*-scores were calculated according to reference values based on height and gender [32].

### 2.6. Statistical Analysis

All statistical analyses were performed using SPSS 20.0 for Windows (SPSS Inc., Chicago, IL, USA). BMI *z*-score, sensor glucose measurements, and cardiometabolic risk parameters at baseline and after 12 months of lifestyle intervention were compared using the paired Student’s *t*-test, the Wilcoxon signed-rank test, or the *χ*^2^ test, as appropriate. Correlations between variables were determined by Spearman’s correlation analysis. Data are presented as means with standard deviations or as medians with the minima and maxima. For all analyses, a *p*-value below 0.05 was considered to be statistically significant.

## 3. Results

### 3.1. Participant Characteristics at Baseline and after 12 Months of Intervention

Thirty-three, predominantly Caucasian (85%) children (13 boys, 20 girls) with a mean age of 12.5 ± 3.2 years were included. At baseline, 9% were overweight (*n* = 3), 42% (*n* = 14) were obese, and 49% (*n* = 16) were morbidly obese. Despite the wide range in BMI *z*-scores, all children had fasting plasma glucose concentrations within the normal range (<5.6 mmol/L). The mean HOMA-IR was 3.31 ± 1.61. Based on these HOMA-IR values, insulin resistance was present in 58% (*n* = 19) of the children. One child was classified as having impaired glucose tolerance. HbA1c concentrations were elevated in 27% (*n* = 9) of the children. After 12 months of lifestyle intervention, the BMI *z*-score did not improve significantly in the complete group (*p* = 0.206), whereas there was a significant improvement in weight status classification (*p* < 0.001), i.e., a shift from morbidly obese to obese, or obese to overweight (Table 1). TC (*p* = 0.018), LDL-C (*p* = 0.025), and HbA1c concentrations (*p* < 0.001) decreased significantly in the whole group after 12 months of lifestyle intervention (Table 1), while insulin concentrations (*p* = 0.014) and HOMA-IR (*p* = 0.014) increased significantly. The characteristics of the children at baseline and after 12 months of lifestyle intervention are presented in Table 1.

Based on changes in BMI *z*-score after 12 months of lifestyle intervention, children were stratified in two groups: (1) children with a decrease in BMI *z*-score and (2) children with an increase in BMI *z*-score. Sixty-one percent (*n* = 20) of the children successfully improved their BMI *z*-score, with a significant decrease of −0.24 ± 0.15 units (*p* < 0.001). BMI *z*-score increased significantly by 0.21 ± 0.17 units (*p* = 0.001) in the remaining 39% (*n* = 13) of the children. Children with a decrease in BMI *z*-score over time were significantly younger at baseline as compared with children that showed an increase in BMI *z*-score. There were no significant differences at baseline between these groups regarding anthropometric measurements and cardiovascular risk parameters (Table 2). Significant improvements in TC and LDL-C were only demonstrated in children with a decrease in BMI *z*-score. HbA1c concentrations improved significantly in both groups (Table 2). A significant increase in fasting insulin concentrations was found in the children with an increase in BMI *z*-score (Table 2).

### 3.2. 48-Hour Glycemic Profile Analysis at Baseline and after 12 Months of Intervention

The median sensor glucose concentration at baseline was 5.0 (3.2–7.3) mmol/L and did not change significantly after 12 months of lifestyle intervention (*p* = 0.431) (Table 1). CONGA1 decreased significantly after 12 months of lifestyle intervention from 0.57 (0.39–1.31) to 0.50 (0.30–1.08) (*p* = 0.048) (Table 1), but CONGA2 and CONGA4 did not change significantly (*p* = 0.091 and *p* = 0.228, respectively). 

At baseline, sixty-four percent (*n* = 21) of the children reached high-normal sensor glucose concentrations, defined as values ≥6.7 mmol/L, during the 48-hour measuring period. Of these children, six out of 21 no longer reached these high-normal sensor glucose concentrations after 12 months of lifestyle intervention. In addition, a significant decrease in the duration of glucose concentrations being ≥6.7 mmol/L (*p* = 0.001) was demonstrated after 12 months of intervention in the complete group (Figure 1 and Appendix A).

At baseline, 36% (*n* = 12) of the children showed sensor glucose concentrations ≥7.8 mmol/L during the 48-hour measuring period. The duration of glucose concentrations exceeding this threshold did not change significantly after 12 months of lifestyle intervention (*p* = 0.408) (Figure 1 and Appendix A). However, in nine out of 12 of the children that exceeded this threshold at baseline, all sensor glucose concentrations remained below 7.8 mmol/L after 12 months of intervention. In both children exceeding glucose concentrations of 10.0 mmol/L (*n* = 2) at baseline, all sensor glucose concentrations were <6.7 mmol/L after 12 months of intervention. Furthermore, changes in the opposite direction were also observed, i.e., eight out of 12 children with glucose concentrations <6.7 mmol/L at baseline exceeded this threshold after 12 months of lifestyle intervention. Four of these children did not show a decrease in BMI *z*-score after the intervention. 

Finally, at the low end of the plasma glucose spectrum, 73% (*n* = 24) of the children reached sensor glucose concentrations below 3.9 mmol/L at baseline. The total duration for hypoglycemic sensor glucose concentrations did not change significantly after 12 months of intervention (*p* = 0.323) (Figure 1 and Table 1).

In the children who demonstrated a decrease in BMI *z*-score, the duration of sensor glucose concentrations being ≥6.7 mmol/L decreased significantly (*p* = 0.009), and in the children who demonstrated an increase in BMI *z* score, there was a trend towards a shorter duration of sensor glucose concentrations ≥6.7 mmol/L (*p* = 0.060) (Table 3). CONGA1 and CONGA2 decreased significantly in children with a decrease in BMI *z*-score (*p* = 0.021; *p* = 0.048, respectively), whereas there were no significant changes found in children with an increase in BMI *z*-score (*p* = 0.552; *p* = 0.650, respectively) (Table 3). Sensor glucose measurements stratified for daytime (i.e., 7 a.m.–10 p.m.) and nighttime (i.e., 10 p.m.–7 a.m.) are presented in Appendix A. The median, minimum, and maximum sensor glucose concentrations; CV; and the duration for which the glucose concentrations were within the different ranges during the daytime and nighttime did not differ significantly at baseline compared to after 12 months of lifestyle intervention. The durations of sensor glucose concentrations being within the different ranges during the daytime and nighttime also did not change significantly after lifestyle intervention after stratification for children with and without a decrease in BMI *z*-score.

### 3.3. Associations between Changes in Glucose Metabolism and CVD Risk after 12 Months of Lifestyle Intervention

After 12 months of lifestyle intervention, the delta of the median sensor glucose concentration showed a positive association with the delta SBP *z*-score (*r* = 0.405, *p* = 0.024) and delta DBP *z*-score (*r* = 0.414, *p* = 0.021) in the whole group. The delta of the median sensor glucose concentration was not associated with the delta of the anthropometric measurements or the other cardiovascular risk parameters. Delta CONGA1, delta CONGA2, delta CONGA4, and delta CV were not associated with alterations in any of the anthropometric measurements or cardiovascular risk parameters in the whole group.

In the children with a decrease in BMI *z*-score, there was a positive association between the delta BMI *z*-score and delta CONGA1 (*r* = 0.601, *p* = 0.005), delta CONGA2 (*r* = 0.643, *p* = 0.002), delta CONGA4 (*r* = 0.686, *p* = 0.001), and delta CV (*r* = 0.620, *p* = 0.004). Moreover, in these children, the delta median sensor glucose was positively associated with the delta LDL-C (*r* = 0.472, *p* = 0.036), delta SBP *z*-score (*r* = 0.598, *p* = 0.005), and delta DBP *z*-score (*r* = 0.605, *p* = 0.005). The delta CV was positively associated with the delta TC (*r* = 0.453, *p* = 0.045) and delta serum triglyceride concentration (*r* = 0.457, *p* = 0.043). In children with an increase in BMI *z*-score, no associations were found between the delta BMI *z*-score and alterations in sensor glucose measurements. In these children, the delta maximum sensor glucose concentration showed inverse associations with delta TC (*r* = −0.581, *p* = 0.047) and delta LDL-C (*r* = −0.580, *p* = 0.048). Correlation coefficients stratified for changes in BMI *z*-score are presented in Table 4 and Appendix A. Further corrected regression analyses were not performed, due to limited statistical power after stratification for increases or decreases in BMI *z*-score.

## 4. Discussion

This is the first study investigating the effect of a long-term lifestyle intervention on glycemic profiles in free-living conditions in children with overweight and with (morbid) obesity. Our results demonstrate that the durations for which glucose concentrations were in the high-normal range and the glycemic variability calculated as CONGA1 decreased significantly after 12 months of lifestyle intervention. Furthermore, the delta of the median glucose concentrations in free-living conditions was positively associated with the deltas of the SBP and DBP *z*-scores. These associations were only present in children with a decrease in BMI *z*-score. Our results suggest that an on-going, tailored, outpatient lifestyle intervention can result in the improvement of glycemic profiles in free-living conditions and that these improvements may coincide with a decreased CVD risk in children with overweight and with (morbid) obesity.

In a previous cross-sectional study in children with overweight and with (morbid) obesity, we demonstrated that glycemic profiles in free-living conditions were aberrant [9]. This was not just the consequence of excess body weight, since none of the sensor glucose measurements were associated with BMI *z*-score [9]. We now show not only that a lifestyle intervention improves glycemic profiles, but also that a reduction in BMI *z*-score coincides with improvements of glycemic profiles. The duration of glucose concentrations being in the high-normal range and the CONGA1 only improved in children with a decrease in BMI *z*-score. Notably, associations between the delta of the median sensor glucose concentrations and deltas of the SBP and DBP *z*-scores were only found in the subgroup of children with a decreased BMI *z*-score. It is tempting to suggest that a decrease in BMI *z*-score is the result of lifestyle improvements. Dietary composition and quality as well as physical activity were important aspects of the lifestyle intervention and are factors well-known to interact with glucose homeostasis [33]. As mentioned previously, in adults with T2DM, it has been demonstrated that interventions targeting diet or physical activity both resulted in a significant improvement of glycemic variability [16,17,18,19]. In contrast to these standardized interventions, our intervention was aimed at gradual lifestyle improvements taking into account the personal needs and opportunities of each family, resulting in a wide heterogeneity of dietary intake and physical activity. Since these factors were not assessed in detail in this study, we cannot differentiate whether the observed positive effects on glycemic profiles are the result of improvement in weight, improvement of lifestyle, or a combination of both. In future studies, it needs to be elucidated which specific modifiable factors contribute to the improvements in glycemic profiles in children with overweight and with (morbid) obesity in free-living conditions, in order to facilitate a better definition of targets for future intervention strategies. Notwithstanding this, these results underscore that an on-going, tailored, outpatient lifestyle intervention resulted in beneficial effects on glucose homeostasis and CVD risk in children with a decrease in BMI *z*-score. 

In the current study, glycemic variability was assessed using the CONGA and the coefficient of variation. In general, high glucose variability is thought to be harmful for peripheral vascular function [11,12,13,14] and brain development in children [15]. Currently, the reference ranges for glycemic variability as estimated with the CONGA1, 2, and 4 in healthy children with a normal weight are unknown. However, the CV in our study was comparable to the values that were described in a previous study in 57 healthy children [34]. Interestingly, the significant improvement in CONGA, as shown in our study, illustrates that improvement of the glycemic variability is possible in children with overweight and with (morbid) obesity via lifestyle adaptations. In addition to CONGA, glycemic control over a longer period of time was evaluated by assessing HbA1c concentrations. In both children demonstrating a decrease and increase in BMI *z*-score, HbA1c concentrations improved significantly after 12 months of lifestyle intervention. By contrast, we found a significant increase in fasting insulin levels in the children with an increase in BMI *z*-score after the intervention. These findings are in line with a previous study in children, in which there was a strong association between an increasing degree of insulin resistance and increasing adiposity [35]. Similarly to our findings, that study also did not find increasing HbA1c concentrations with increasing degrees of adiposity in Caucasian European children. They did find an association between HbA1c and adiposity in children of South Asian and African American origin [35]. These findings imply that ethnic differences play a role in the association between adiposity and glucose homeostasis, but also that other factors besides adiposity contribute to HbA1c concentrations. Previous studies in adults with T2DM demonstrated that the control of postprandial hyperglycemia is a very important contributor to HbA1c concentrations [36,37]. Taking into account the decreased duration for which glucose concentrations were high-normal, the improvement of CONGA1 and 2, and the improvement of HbA1c concentrations, we hypothesize that this improvement in glucose homeostasis might be due to a reduction in postprandial glucose excursions after 12 months of lifestyle intervention. Although a subgroup of the children did not have a decrease in BMI *z*-score, lifestyle changes made during the intervention might have positively affected HbA1c concentrations despite an increase in BMI *z*-score.

The exact underlying mechanisms and sequence of events resulting in glucose dysregulation are not fully understood, but there is strong evidence suggesting a link between glucose dysregulation and dyslipidemia [38]. In this study, the delta maximum sensor glucose concentration was inversely associated with delta TC and delta LDL-C in the children with an increase in BMI *z*-score. Furthermore, it was shown that the delta of the median glucose concentration was associated with the deltas of the SBP and DBP *z*-scores, only in children with a decrease in BMI *z*-score. Interestingly, these correlations between the delta of the median glucose concentrations and deltas of SBP and DBP *z*-scores were not found for the deltas of the CV and CONGA1, CONGA2, or CONGA4. In children with an increase in BMI *z*-score, HbA1c concentrations improved significantly, while cardiovascular risk parameters showed no significant improvements. These results suggest that changes in glucose homeostasis may, but do not necessarily, coincide with changes in cardiovascular risk parameters.

Due to the long-term follow-up of our intervention, we did not include a control group with random assignment of treatment, because it was not ethically justifiable to keep children in a control program for a prolonged period of time and withhold treatment from them. This can be considered as a limitation of this study. Furthermore, the cohort size of our study might seem small, and therefore the affirmation of our findings in larger cohort studies is certainly recommendable. However, considering the current literature and the novelty of investigating sensor glucose measurements in children without a diagnosis of T2DM, it can be argued that we have a relevant study population size and that the results of this study might create awareness that further research is needed. It would have been valuable if healthy children with a normal weight were included in this study as a reference population for the normality of glycemic profiles, since the current evidence in this population is limited [34,39]. Additionally, it would be interesting to investigate which modifiable factors contribute to glycemic profiles in free-living conditions in children with overweight and with (morbid) obesity, for example, by objectively assessing physical activity using an accelerometer.

## 5. Conclusions

In children with a decrease in BMI *z*-score, glycemic profiles in free-living conditions improve after 12 months of lifestyle intervention, as demonstrated by a decrease in the duration of time that glucose concentrations are in the high-normal range and by the decrease in CONGA1. Changes in median glucose concentrations are associated with changes in SBP and DBP *z*-scores in children with overweight and with (morbid) obesity, but only in those who showed a decrease in BMI *z*-score. These results suggest that a lifestyle intervention can result in the improvement of glucose homeostasis and that these improvements are also associated with improvements in cardiovascular health parameters. Next, long-term follow-up studies are necessary to evaluate whether the improvement of glycemic profiles in free-living conditions during childhood results in long-term health benefits.

## Figures and Tables

**Figure 1 nutrients-12-01228-f001:**
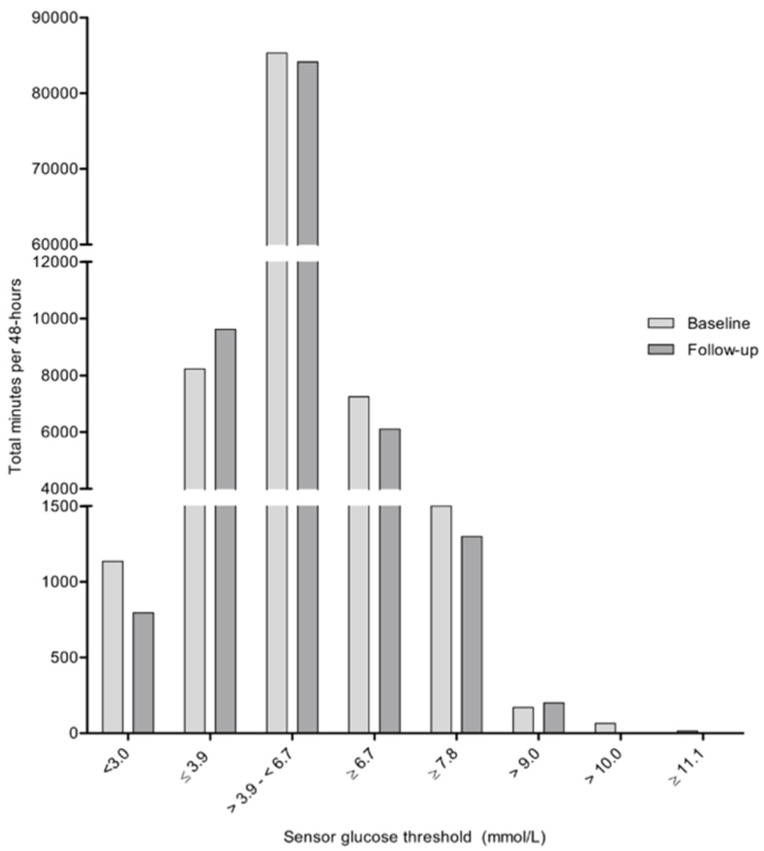
Total minutes per 48 hours that sensor glucose concentrations were within specific glucose thresholds for the whole study population, at baseline and after 12 months of lifestyle intervention.

**Table 1 nutrients-12-01228-t001:** Characteristics of the study participants at baseline and after 12 months of lifestyle intervention.

Characteristics	Baseline	After 12 Months of Intervention
Age, years	12.5 ± 3.2	13.8 ± 3.0
BMI *z*-score	3.53 ± 0.66	3.46 ± 0.67
Overweight/obesity/morbid obesity, %	9/42/49	15/39/46 *
Waist circumference *z*-score	6.85 ± 2.43	7.33 ± 2.12 *
Median sensor glucose, mmol/L	5.0 (3.2–7.3)	5.1 (3.6–6.9)
Maximum sensor glucose, mmol/L	7.2 (5.6–11.2)	7.0 (5.4–9.9)
Minimum sensor glucose, mmol/L	3.4 (2.2–4.4)	3.4 (2.2–4.9)
Sensor glucose area under the curve	14867 ± 1447	14746 ± 1586
CONGA1	0.57 (0.39–1.31)	0.50 (0.30–1.08) *
CONGA2	0.72 (0.46–1.61)	0.69 (0.30–1.58)
CONGA4	0.88 (0.45–2.02)	0.87 (0.39–1.94)
CV, %	15.7 ± 5.5	15.1 ± 4.0
Fasting glucose, mmol/L	4.0 ± 0.5	4.0 ± 0.5
Fasting insulin, mU/L	18.5 ± 9.2	25.6 ± 13.7 *
HOMA-IR	3.31 ± 1.61	4.29 ± 2.30 *
HbA1c, %	5.4 **±** 0.3	5.2 ± 0.4 *
Plasma glucose 2 hours after glucose load, mmol/L	5.5 ± 1.2	5.6 ± 1.1
Total cholesterol, mmol/L	4.8 (3.5–6.6)	4.5 (3.5–6.9) *
LDL-cholesterol, mmol/L	3.1 (2.0–4.5)	2.7 (1.7–4.6) *
HDL-cholesterol, mmol/L	1.1 (0.8–1.9)	1.1 (0.8–1.9)
Triglycerides, mmol/L	1.21 (0.39–4.48)	1.13 (0.51–3.77)
Systolic blood pressure *z*-score	0.19 ± 1.26	0.02 ± 1.16
Diastolic blood pressure *z*-score	−0.37 ± 0.88	−0.66 ± 1.17

Data are presented as mean ± SD or as median (minimum–maximum). * = significant between parameters at baseline compared to parameters after 12 months of intervention. CONGA = continuous overlapping net glycemic action; CONGA presented for 1-, 2-, or 4-hour time differences; CV = coefficient of variation; HOMA-IR = Homeostatic Model Assessment of Insulin Resistance.

**Table 2 nutrients-12-01228-t002:** Characteristics of the study participants at baseline and after 12 months of lifestyle intervention stratified for change in BMI *z*-score.

	Decrease in BMI *z*-Score after 12 Months of Intervention (*n* = 20)		Increase in BMI *z*-Score after 12 Months of Intervention (*n* = 13)	
	Baseline	12 Months of Intervention	Baseline	12 Months of Intervention
Age	11.3 ± 2.6	12.7 ± 2.5	14.4 ± 3.1	15.6 ± 3.0
BMI *z*-score	3.54 ± 0.62	3.30 ± 0.61 *	3.50 ± 0.74	3.71 ± 0.71 *
Waist circumference *z*-score	6.3 ± 1.9	6.9 ± 1.6	7.6 ± 3.4	8.2 ±3.1
Fasting glucose, mmol/L	4.1 ± 0.6	4.3 ± 0.4	3.9 ± 0.5	4.1 ± 0.6
Fasting insulin, mU/L	18.1 ± 7.0	20.7 ± 9.8	20.2 ± 12.4	27.5 ± 13.3 *
HOMA-IR	3.21 ± 1.40	4.00 ± 2.00	3.20 ± 1.84	4.50 ± 2.48
HbA1c, %	5.4 ± 0.3	5.2 ± 0.4 *	5.5 ± 0.2	5.2 ± 0.4 *
Plasma glucose 2 hours after glucose load, mmol/L	5.6 (3.9–7.5)	5.6 (4.2–6.7)	4.8 (2.9–6.3)	5.2 (4.2–8.5)
Total cholesterol, mmol/L	5.0 ± 0.9	4.4 ± 0.6 *	4.9 ± 0.7	5.0 ± 0.9
LDL-cholesterol, mmol/L	3.2 ± 0.8	2.7 ± 0.6 *	2.9 ± 0.7	3.0 ± 0.7
HDL-cholesterol, mmol/L	1.1 ± 0.2	1.2 ± 0.2	1.3 ± 0.4	1.2 ± 0.3
Triglycerides, mmol/L	1.23 (0.49–3.11)	1.02 (0.51–3.69)	1.20 (0.39–4.48)	1.28 (0.65–3.77)
Systolic blood pressure *z*-score	0.23 ± 1.09	−0.11 ± 1.07	0.11 ± 1.63	0.21 ± 1.37
Diastolic blood pressure *z*-score	−0.55 ± 0.65	−0.89 ± 1.23	−0.17 ± 1.15	−0.16 ± 0.95

Data are presented as mean ± SD or as median (minimum–maximum). * = significant difference between parameters at baseline compared to parameters after 12 months of intervention. HOMA-IR = homeostatic model assessment of insulin resistance.

**Table 3 nutrients-12-01228-t003:** The 48-hour sensor glucose measurements stratified for changes in BMI *z*-score.

	Decrease in BMI *z*-Score after 12 Months of Intervention (*n* = 20)		Increase in BMI *z*-score after 12 Months of Intervention (*n* = 13)	
	Baseline	After 12 Months of Intervention	Baseline	After 12 Months of Intervention
Median sensor glucose, mmol/L	4.9 (3.2–7.3)	4.9 (3.6–5.9)	5.0 (3.7–6.1)	5.1 (3.6–6.9)
Maximum sensor glucose, mmol/L	7.0 (6.0–9.5)	6.9 (5.4–8.7)	7.5 (5.6–11.2)	7.3 (5.9–9.9)
Minimum sensor glucose, mmol/L	3.4 (2.2–4.4)	3.3 (2.6–4.8)	3.3 (2.2–3.9)	3.6 (2.2–4.9)
Time in level 2 hypoglycemic range (<3.0 mmol/L), minutes	0 (0–265)	0 (0–250)	0 (0–395)	0 (0–135)
Time in level 1 hypoglycemic range (3.0–3.9 mmol/L), minutes	58 (0–690)	0 (0–650)	85 (0–870)	45 (0–1270)
Time in target range (3.9–10.0 mmol/L), minutes	2823 (2050–2880)	2710 (2195–2880)	2795 (1615–2880)	2835 (1545–2880)
Time in secondary target range (3.9–7.8 mmol/L), minutes	2735 (1945–2880)	2703 (2195–2880)	2655 (1615–2880)	2560 (1545–2880)
Time in high-normal range (≥6.7 mmol/L), minutes	38 (0–895)	7 (0–123) *	205 (0–840)	22 (0–210)
Time in high-normal range (≥7.8 mmol/L), minutes	0 (0–190)	0 (0–170)	0 (0–185)	0 (0–345)
Time in level 1 hyperglycemic range (>10.0 mmol/L), minutes	0 (0–0)	0 (0–0)	0 (0–40)	0 (0–0)
Time in level 2 hyperglycemic range (>13.9 mmol/L), minutes	0 (0–0)	0 (0–0)	0 (0–0)	0 (0–0)
Sensor glucose area under the curve	14729 ± 1214	14419 ± 1301	15080 ± 1780	15249 ± 1891
CV, %	15.1 ± 5.4	14.5 ± 2.7	16.7 ± 5.6	16.1 ± 5.3
CONGA1	0.56 (0.39–1.00)	0.49 (0.30–1.00) *	0.65 (0.39–1.31)	0.60 (0.30–1.08)
CONGA2	0.70 (0.46–1.26)	0.63 (0.39–1.16) *	0.77 (0.46–1.61)	0.71 (0.30–1.58)
CONGA4	0.83 (0.45–1.51)	0.87 (0.48–1.31)	0.95 (0.66–2.02)	0.80 (0.39–1.94)

Data are presented as mean ± SD or as median (minimum–maximum). * = significant difference between baseline and after 12 months of lifestyle intervention at the 0.05 level. CV = coefficient of variation; CONGA = continuous overlapping net glycemic action; CONGA presented for 1-, 2-, or 4-hour time differences.

**Table 4 nutrients-12-01228-t004:** Correlation coefficients between baseline characteristics and sensor glucose measurements—subgroup analysis for the children with a decrease in BMI *z*-score.

	Δ Median sensor glucose	Δ Maximum sensor glucose	Δ Minimum sensor glucose	Δ CONGA1	Δ CONGA2	Δ CONGA4	Δ CV	Δ AUC
Δ BMI *z*-score	−0.127	0.502 *	−0.356	0.601 *	0.643 *	0.686 *	0.620 *	0.214
Δ Fasting glucose	−0.067	−0.241	−0.342	−0.095	−0.076	0.03	0.350	−0.416
Δ Fasting insulin	0.150	−0.183	−0.05	−0.084	0.082	0.11	0.000	0.075
Δ HOMA-IR	−0.072	−0.034	−0.023	−0.179	0.216	0.191	0.197	−0.038
Δ HbA1c	0.340	0.153	−0.175	0.340	0.319	0.239	0.032	0.229
Δ Glucose 2 h after glucose load	0.027	0.144	−0.412	0.155	0.236	0.292	0.490	−0.259
Δ Total cholesterol	0.305	0.342	−0.196	0.170	0.207	0.282	0.453 *	0.162
Δ LDL-cholesterol	0.472 *	0.150	0.029	0.166	0.045	0.021	0.111	0.214
Δ HDL-cholesterol	0.041	0.065	−0.167	−0.214	−0.061	0.108	0.172	0.056
Δ Triglycerides	−0.203	0.404	−0.445 *	0.228	0.433	0.520 *	0.457 *	−0.117
Δ Systolic blood pressure *z*-score	0.598 *	0.268	0.095	−0.033	0.019	0.020	−0.104	0.498 *
Δ Diastolic blood pressure z-score	0.605 *	0.148	0.140	0.052	0.010	−0.093	−0.335	0.366

Correlations between variables were determined by Pearson’s correlation coefficient or Spearman’s correlation analysis, as appropriate, and were not corrected for other variables. * = significant correlation. Δ= delta; HOMA-IR = homeostatic model assessment of insulin resistance; CV = coefficient of variation; CONGA = continuous overlapping net glycemic action; CONGA presented for 1-, 2-, or 4-hour time differences; AUC = area under the curve.

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
