# Peer review of "Changes in Free-Living Glycemic Profiles after 12 Months of Lifestyle Intervention in Children with Overweight and with Obesity"

_nutrients, 2020, doi:10.3390/nu12051228_

Round 1
Reviewer 1 Report
Karnebeek et al have evaluated the effect of 12 months lifestyle intervention on free-living glycemic profiles in non-diabetic children with overweight and obesity in their manuscript entitled
“Changes in free-living glycemic profiles after 12 months of lifestyle intervention in children with overweight and obesity” They evaluated 33 children children and conclude that
the glycemic profiles in free-living conditions in children who were overweight improved in children with a decrease in BMI z-score after lifestyle intervention. Additionally, in this sub-group changes in median sensor glucose concentrations were associated with changes in LDL-cholesterol and blood pressure z scores.
Overall, the manuscript is well-written and clear and studies like this undertaken to evaluate interventions in real world situations are of significant importance.
However, some of the claims are overstated and need further clarification for the ease of the reader.
To improve readability of the manuscript could the authors please define the International Obesity Task Force criteria for overweight, obese, or morbidly obese within the text.
The end sentence of the abstract describing cardiovascular health is an overstatement please alter to be more relevant to the data presented. “
In the Introduction a great outline of what constitutes cardiometabolic risk profiles would provide relevance for the variables chosen
“durable lifestyle changes” should be explained more clearly as although referenced #5 this will be of interest to readers.
Can the authors speculate in the discussion why insulin levels are significantly elevated in the 13 people who had an increase in BMI ?
Author Response
Karnebeek et al have evaluated the effect of 12 months lifestyle intervention on free-living glycemic profiles in non-diabetic children with overweight and obesity in their manuscript entitled
“Changes in free-living glycemic profiles after 12 months of lifestyle intervention in children with overweight and obesity” They evaluated 33 children children and conclude that
the glycemic profiles in free-living conditions in children who were overweight improved in children with a decrease in BMI z-score after lifestyle intervention. Additionally, in this sub-group changes in median sensor glucose concentrations were associated with changes in LDL-cholesterol and blood pressure z scores.
Overall, the manuscript is well-written and clear and studies like this undertaken to evaluate interventions in real world situations are of significant importance.
However, some of the claims are overstated and need further clarification for the ease of the reader.
1) To improve readability of the manuscript could the authors please define the International Obesity Task Force criteria for overweight, obese, or morbidly obese within the text.
Response 1) We have elaborated further on the International Obesity Taskforce criteria in the Methods section of the manuscript (page 3, section 2.3 Patients characteristics).
2) The end sentence of the abstract describing cardiovascular health is an overstatement please alter to be more relevant to the data presented. “
Response 2) We have adapted the last sentence of the abstract to more accurately reflect the findings presented in this manuscript.
3) In the Introduction a great outline of what constitutes cardiometabolic risk profiles would provide relevance for the variables chosen
Response 3) We have added examples of which cardiometabolic parameters were shown to be aberrant in previous studies in children with obesity and were also shown to be important predictors of atherosclerotic cardiovascular disease in adulthood.
4) “durable lifestyle changes” should be explained more clearly as although referenced #5 this will be of interest to readers.
Response 4) We have elaborated on how the intervention aimed to achieve durable lifestyle changes in paragraph 2.1 Setting of the Methods section, page 2).
5) Can the authors speculate in the discussion why insulin levels are significantly elevated in the 13 people who had an increase in BMI ?
Response 5) These findings are in line with a previous study in children, in which there was a strong association between an increasing degree of insulin resistance and increasing adiposity (Nightingale et al, Diabetes Care 2013 Jun; 36(6): 1712-1719). We have commented on this in the discussion section of the manuscript (page 10).
Reviewer 2 Report
- Throughout the manuscript, add sample source of glucose concentration and triacylglycerol concentration.
- The manuscript would benefit from review by a native English speaker who also have extensive knowledge in scientific writing.
- Lines 109-111, add how fasting plans glucose conc, serum insulin conc, and HbA1c conc were determined.
- Tables: use *, **, *** to indicate significant differences in parameters between baseline and after 12 months intervention.
- Table 2: add age
- Line 189, a significant increase in HOMA-IR was not indicated in Table 2.
- Figure 1. Add standard deviations.
- Figure 1. Check significances in sensor glucose conc <3.0 and <-3.9.
- Table 3, Check significances in time in high normal range, CONGA2, CONGA4 between baseline and after 12 months intervention in children who showed an increase in BMI-z score.
- Table, list confounders.
- Table 2, in children showing an increase in BMI z-score improved HbA1c. However they also increased fasting insulin level and insulin resistance measured by HOMA-IR. It should be discussed in the discussion.
Author Response
1) Throughout the manuscript, add sample source of glucose concentration and triacylglycerol concentration. Response 1) We have added the sample source of glucose and triglyceride concentrations throughout the manuscript.
2) The manuscript would benefit from review by a native English speaker who also have extensive knowledge in scientific writing.
Response 2) We have made some grammatical changes throughout the manuscript.
3) Lines 109-111, add how fasting plans glucose conc, serum insulin conc, and HbA1c conc were determined.
Response 3) Analysis methods were added to the methods section of the manuscript, paragraphs 2.4 and 2.5.
4) Tables: use *, **, *** to indicate significant differences in parameters between baseline and after 12 months intervention.
Response 4) * were used to indicate significant differences between parameters at baseline and after the intervention in all tables.
5) Table 2: add age
Response 5) We have added age in tables 1 and 2.
6) Line 189, a significant increase in HOMA-IR was not indicated in Table 2.
Response 6) The reviewer is correct that a significant increase in HOMA-IR was not indicated in table 2. We have mistakenly written that there was a significant increase in HOMA-IR and fasting insulin concentrations in children with an increase in BMI z-score. However, statistical significance was reached only for difference in fasting insulin concentrations, but not for HOMA-IR (p=0.091). We have corrected this in the manuscript in the last sentence of paragraph 3.1.
7) Figure 1. Add standard deviations.
Response 7) Figure 1 reflects the total number of minutes that glucose concentrations were within the different ranges for the whole study population (i.e. the sum of all minutes within a certain ranges of all participants). Therefore there are no standard deviations to be displayed.
This figure is an addition to Supplemental tables 1 and 2, and Table 3 in the main manuscript, where we have presented the time within the different glucose ranges as median and range of the population (rather than the sum).
8) Figure 1. Check significances in sensor glucose conc <3.0 and <-3.9.
Response 8) As we have shown in Supplemental table 1, the number of minutes that glucose ranges were <3.0 mmol/L or <3.9 mmol/L, did not differ significantly between baseline and after one year of intervention.
Since Figure 1 displays the sum of all minutes within the different ranges of all participants, we recognize that it can be confusing that we have presented a statistical difference in time glucose concentrations above 6.7 mmol/L in this figure, in addition to the tables. We have therefor removed the * from figure 1.
9) Table 3, Check significances in time in high normal range, CONGA2, CONGA4 between baseline and after 12 months intervention in children who showed an increase in BMI-z score.
Response 9) Differences in time in high normal range, CONGA2 (p=0.650) and CONGA4 (p=0.600) between baseline and after one year of intervention did not reach statistical significance.
10) Table, list confounders.
Response 10) The correlations shown in table 4 were not corrected for confounders. We have added a statement about this in the footnote of the table.
11) Table 2, in children showing an increase in BMI z-score improved HbA1c. However they also increased fasting insulin level and insulin resistance measured by HOMA-IR. It should be discussed in the discussion.
Response 11) We indeed found a significant increase in fasting insulin levels in the children with an increase in BMI z-score after the intervention. These findings are in line with a previous study in children, in which there was a strong association between an increasing degree of insulin resistance and increasing adiposity [33]. Similar to our findings, that study also did not find increasing HbA1c concentrations with increasing degrees of adiposity in Caucasian European children. They did find an association between HbA1c and adiposity in children of South Asian and African American origin. We have commented on this in the discussion section of the manuscript (page 10).
Round 2
Reviewer 2 Report
For table 4, explain why they were not corrected for confounders.
For the last comment, in children showing an increase in BMI z-score improved HbA1c. However they also increased fasting insulin level and insulin resistance measured by HOMA-IR,
Did the study[33] report "improving" HbA1c? Also, did you report the association between adiposity and HbA1c? What is your justification for in children showing an increase in BMI z-score "improved" HbA1c AND increased fasting insulin level and insulin resistance?
Author Response
1) For table 4, explain why they were not corrected for confounders.
Response 1) We agree that it can be interesting to further assess the correlations from table 4 with corrected regression analyses. However, in order to ensure enough statistical power for reliable regression analysis, we believe a larger sample size would be necessary. Therefore, corrected analyses were not added to this manuscript. We have added a sentence to clarify this in the manuscript (page 9).
2) For the last comment, in children showing an increase in BMI z-score improved HbA1c. However they also increased fasting insulin level and insulin resistance measured by HOMA-IR,
Did the study[33] report "improving" HbA1c? Also, did you report the association between adiposity and HbA1c? What is your justification for in children showing an increase in BMI z-score "improved" HbA1c AND increased fasting insulin level and insulin resistance?
Response 2) The study that we referred to regarding the association between HbA1c and adiposity is a cross-sectional study and did therefor not report on improving HbA1c. In their cross-sectional analyses did not find increasing HbA1c concentrations with increasing degrees of adiposity in Caucasian European children. They did find a strong association between an increasing degree of fasting insulin resistance and increasing adiposity.
In our study, there was no significant association between changes in HbA1c and changes in BMI z-score in the children with an increase in BMI z-score. This finding, combined with the results from the study referred to above, suggest that additional parameters besides obesity play an important role in determining HbA1c concentrations in children (and that contributing factors might be different for insulin concentrations). For instance, previous studies in adults with T2DM demonstrated that control of postprandial hyperglycemia is a very important contributor to HbA1c concentrations (refs 36 and 37 of the manuscript). In our study, glucose concentrations in the high-normal range and glycemic variability also showed decreasing trends (although not a significant decrease) in the children with an increase in BMI z-score. These changes are hypothesized to contribute to an improvement of HbA1c.
Although this subgroup of children did not decrease their BMI z-score, they still made some lifestyle changes that could have positively influenced HbA1c concentrations. We have made some additions to this paragraph in the discussion section of the manuscript.